# Topical Delivery of Diacetyl Boldine in a Microemulsion Formulation for Chemoprotection against Melanoma

**DOI:** 10.3390/pharmaceutics15030901

**Published:** 2023-03-10

**Authors:** Ahmed Al Saqr, Manjusha Annaji, Ishwor Poudel, Mohammed F. Aldawsari, Hamad Alrbyawi, Nur Mita, Muralikrishnan Dhanasekaran, Sai H. S. Boddu, Rabin Neupane, Amit K. Tiwari, R. Jayachandra Babu

**Affiliations:** 1Department of Drug Discovery and Development, Harrison School of Pharmacy, Auburn University, Auburn, AL 36849, USA; 2Department of Pharmaceutics, College of Pharmacy, Prince Sattam Bin Abdulaziz University, Al-kharj 11942, Saudi Arabia; 3Pharmaceutics and Pharmaceutical Technology Department, College of Pharmacy, Taibah University, Medina 42353, Saudi Arabia; 4Department of Pharmaceutical Sciences, College of Pharmacy and Health Sciences, Ajman University, Ajman P.O. Box 346, United Arab Emirates; 5Center of Medical and Bio-Allied Health Sciences Research, Ajman University, Ajman P.O. Box 346, United Arab Emirates; 6Department of Pharmacology and Experimental Therapeutics, The University of Toledo, Toledo, OH 43614, USA; 7Department of Cell and Cancer Biology, College of Medicine and Life Sciences, University of Toledo, Toledo, OH 43614, USA

**Keywords:** diacetyl boldine, melanoma, microemulsion

## Abstract

This study aimed to develop a microemulsion formulation for topical delivery of Diacetyl Boldine (DAB) and to evaluate its cytotoxicity against melanoma cell line (B16BL6) in vitro. Using a pseudo-ternary phase diagram, the optimal microemulsion formulation region was identified, and its particle size, viscosity, pH, and in vitro release characteristics were determined. Permeation studies were performed on excised human skin using Franz diffusion cell assembly. The cytotoxicity of the formulations on B16BL6 melanoma cell lines was evaluated by MTT (3-(4,5-dimethylthiazol-2-yl)-2,5-diphenyl-2H-tetrazolium bromide) assay. Two formulation compositions were selected based on the higher microemulsion area of the pseudo-ternary phase diagrams. The formulations showed a mean globule size of around 50 nm and a polydispersity index of <0.2. The ex vivo skin permeation study demonstrated that the microemulsion formulation exhibited significantly higher skin retention levels than the DAB solution in MCT oil (Control, DAB-MCT). Furthermore, the formulations showed substantially higher cytotoxicity toward B16BL6 cell lines than the control formulation (*p* < 0.001). The half-maximal inhibitory concentrations (IC50) of F1, F2, and DAB-MCT formulations against B16BL6 cells were calculated to be 1 µg/mL, 10 µg/mL, and 50 µg/mL, respectively. By comparison, the IC50 of F1 was 50-fold lower than that of the DAB-MCT formulation. The results of the present study suggest that microemulsion could be a promising formulation for the topical administration of DAB.

## 1. Introduction

Melanoma is a skin cancer affecting melanocytes (melanin pigment cells) located in the epidermal basal layer. It is the most aggressive form of all skin cancers, with high mortality. Even though melanoma incidence is 1% among skin cancers, it accounts for 80% of all skin cancer deaths. In recent years, melanoma incidence has increased more than any other cancer, with an approximate doubling of rates every 10–20 years. It is estimated that 1 in 38 white Americans develop melanoma, with about 99,780 new cases and nearly 7650 expected deaths in 2022 [1]. Melanoma has become increasingly common in many developed countries among fair-skin populations over the past decades [2]. Dacarbazine is the first chemotherapy drug approved by the US Food and Drug Administration (FDA) for the treatment of melanoma. Currently, other chemotherapy drugs, such as cisplatin, fotemustine, paclitaxel, and vinblastine also can be used to treat melanoma, but most of these drugs induce severe side effects [3]. Therefore, alternative approaches such as target/site-specific drug delivery systems are needed. These systems act locally and provide minimal systemic exposures, thereby minimizing the adverse effects.

Natural products from medicinal plants have been used to treat and prevent several diseases. Those medicinal plants are composed of biologically active compounds such as flavonoids and alkaloids responsible for their effectiveness in treating and preventing some diseases [4]. The anticancer potency of those natural products has garnered substantial attention in the pharmaceutical field for decades. Approximately 60% of commercially available anticancer drugs are plant-derived and from natural resources [5,6].

Boldine is an aporphine alkaloid derivative extracted from the leaves and bark of the Chilean native Boldo plant (*Peumus Boldus*). Recent investigations on boldine have shown potent anticancer effects against breast, hepatic, bladder, and glial cancer cells [7,8,9,10]. In addition to its anticancer and antioxidant effects, it prevents UV-induced skin damage [11]. A clinical study conducted in a group of Caucasians and Asians confirmed the strong inhibition of melanogenesis after treatment with topical 4% DAB [12]. Boldine acts by inhibiting tyrosinase activation by a dual effect with alpha-adrenergic antagonist properties and interference on calcium influx. Furthermore, boldine showed an anti-melanogenic effect with in vitro melanocyte monoculture and 3D spheroid model [13]. Even the bioactive oligopeptides related to boldine were effective in treating skin diseases like melisma compared to 4% hydroquinone on normal skin [14].

Diacetyl boldine (DAB) has a molecular weight of 411.4 g/mol and log P of 2.9 [15]. DAB has been reported to be more effective in treating melasma [16] and hyperpigmentation [17]. The biological fate of DAB and topical permeability has not been explored. However, its parent alkaloid boldine exhibits a short half-life (30 min) and low oral bioavailability due to extensive first-pass metabolism [18]. We postulate that topical administration is an attractive route for delivering DAB as a promising skin-lightening and chemo-protective agent. The advantages of transdermal and topical drug delivery systems include avoiding hepatic metabolism, enhancing the drug’s bioavailability, and reducing inter-patient differences in gastrointestinal tract absorption.

The microemulsion is a promising drug delivery system for both lipophilic and hydrophilic drugs. The overall drug bioavailability and long-term release of many naturally derived phytochemicals can be improved by microemulsion formulation [19,20,21,22]. A microemulsion is spontaneously formed from a composition containing oil, water, surfactant, and co-surfactant. They are transparent and optically isotropic. The droplet size is usually 10–100 nm [23]. Microemulsions have attracted much attention as a topical/transdermal delivery system for several reasons; they are thermodynamically stable, easy to prepare, may enhance the penetration of drug through the skin, and can solubilize poorly water-soluble drugs. Factors such as increased skin hydration, nano-droplet size, and enhanced surface area can affect the permeation of drugs presented in microemulsions. Further, oils or surfactants of the microemulsion could disrupt the lipid structure of the stratum corneum or increase drug solubility in the stratum corneum. Microemulsions also have a high drug-loading capacity which could eventually improve penetration by providing a higher concentration gradient to increase the driving force across the skin [24,25,26,27].

Despite established skin protective, anti-melanogenic effects and cytotoxicity of boldine or DAB against various cancer cell lines, their anti-melanoma effect is yet to be investigated. Besides, there are limited published studies on the topical delivery of DAB across human or animal skin [13,16]. In this study, we designed a DAB microemulsion formulation for enhanced skin permeation and retention across dermatomed human skin. We further evaluated the in vitro cytotoxicity of the microemulsion against B16BL6 melanoma cells.

## 2. Materials and Methods

### 2.1. Materials

DAB was obtained from South American Pharmaceutical (Santiago, Chile). Phosphatidylcholine (Lecithin) was purchased from Spectrum Chemical Corporation (Gardena, CA, USA). Solutol^®^ HS 15 (Kolliphor^®^ HS 15) was purchased from BASF Pharma (Ludwigshafen, Germany). Miglyol^®^ 810N (medium-chain triglycerides) was purchased from Sasol GmbH (Hamburg, Germany) and super-refined propylene glycol was obtained as a gift sample from Croda Inc. (Princeton, NJ, USA). Ethanol USP was purchased from Letco Medical (Decatur, AL, USA). Formic acid and acetonitrile were purchased from EMD Chemical Inc. (Gibbstown, NJ, USA). All other reagents were purchased from VWR International (West Chester, PA, USA). Fetal bovine serum, Dulbecco’s Modified Eagle’s Medium, and other reagents for cell culture were purchased from Mediatech (Manassa, VA, USA). A mouse melanoma cell line (B16BL6) was obtained from the National Cancer Institute (Frederick, MD, USA). Dermatomed human skin (thickness: 0.5 mm) was obtained from Sciencecare (Phoenix, AZ, USA). It was collected from a single donor within eight h of death and frozen at −80 °C until use.

### 2.2. Construction of Pseudo Ternary Phase Diagram and Preparation of Microemulsions

The aqueous titration method was used to construct ternary phase diagrams containing the oil, surfactant, and aqueous phases. A mixture of surfactants was dissolved in the oil phase in various ratios in glass vials. Each mixture was titrated dropwise with water with continuous stirring at 25 °C until it turned turbid. The percentage composition of each component was determined and plotted on triangular coordinates to construct the phase diagrams (Origin software, OriginLab Corp., Northampton, MA, USA).

### 2.3. Preparation and Characterization of Microemulsions

DAB was dissolved in the medium-chain triglyceride (MCT) oil in shell glass vials under magnetic stirring. Then, lecithin, Solutol^®^ HS 15, and propylene glycol were dissolved in the oil at 40 °C. Finally, water was added and mixed well under stirring for 30 min to obtain clear microemulsion formulations. Placebo (no DAB) formulations were prepared by the same procedure without adding DAB.

The pH of the microemulsion formulations was measured using an Accumet Excel XL15 pH Meter (Fisher Scientific, Suwannee, GA, USA). The mean particle size (d50) and size distribution (polydispersity index) of the microemulsion formulations were determined by the light scattering method using Zetasizer Nano ZS (ZEN 3600, Malvern Instruments Ltd., Malvern, Worcestershire, UK). The mean particle size and polydispersity index of the formulations after appropriate dilutions were determined. The viscosity of the formulations was measured using a Brookfield DV-II PCP pro viscometer (Brookfield Engineering Labs Inc., Middleboro, MA, USA). To determine the viscosity (mPa-s), 0.5 mL of the sample was taken in the cone-cup chamber, and spindle 40 z was used at 25 °C.

### 2.4. In Vitro Release of DAB from Microemulsions

The release of DAB from the microemulsion formulation was performed using Franz diffusion cells (PermeGear, Bethlehem, PA, USA). The dialysis membrane (Regenerated cellulose, molecular weight cut off 14,000 Da, Fisher Scientific Suwanne, GA, USA) was soaked in water for 2 h and mounted between the receptor and donor chambers of the diffusion cell apparatus. The cells had a diffusion surface area of 0.64 cm^2^. The receptor chamber was filled with 5 mL of phosphate buffered saline (pH 7.4). The donor chamber was filled with 0.5 mL of the microemulsion formulation. The diffusion cells were maintained at 37 ± 0.5 °C and stirred with magnetic beads at 600 rpm. Samples (1 mL) were withdrawn from the receptor cell at 1, 2, 4, 6, 8, 12, and 24 h and replenished with new-release media. A correction factor was applied to account for the drug removed due to sampling. The samples were quantified using a High-Performance Liquid Chromatography (HPLC) method to be described later. The experiments were conducted in triplicate for each formulation. The DAB release data was fitted into by various kinetic models such as zero-order, first-order, Higuchi’s, Hixson–Crowell, Korsmeyer–Peppas models. Microsoft Excel 365 was used to perform the curve fitting and statistical analysis to find the best fit.

### 2.5. Skin Permeation

For skin permeation studies, the dialysis membrane was replaced by the human cadaver skin. The dermatomed human skin (thickness: 0.5 mm) was obtained from Sciencecare (Phoenix, AZ, USA). It was collected from a single donor within 8 h of death and frozen in 50% glycerol in normal saline at −80 °C until use. The frozen skin was thawed at room temperature for about 20 min before skin permeation experiments. The skin was washed and thoroughly rinsed with phosphate-buffered saline pH 7.4. The skin permeation studies were conducted using dermatomed skin and Franz diffusion cell method as described in the “in vitro release” studies replacing the dialysis membrane with the skin.

### 2.6. Data Analysis

The cumulative amount of DAB that diffused through the membrane and the skin was calculated as follows:$$Cum_{n}=V_{R}\cdot \;C_{n}+V_{col.}\sum_{i=1}^{n-1}C_{i}$$
where, Cum_n_ is the cumulative amount of DAB in the receptor medium, V_R_ is the receptor volume; C_n_ is the concentration of the nth sample, V_col._ is the collected sample volume, and C_i_ is the concentration of the ith sample.

Diffusion parameters were calculated by plotting the cumulative amount permeated in 24 h versus time. The slope of the curve was the flux (J), and the x-intercept of the straight line was the lag time [28].

### 2.7. Skin Retention

Following the 24 h permeation study, the residual formulation on the skin’s surface was removed using a dropper pipette, and the skin was rinsed with phosphate buffered saline using cotton swabs. The active diffusion area of the skin was cut, weighed, minced, and placed in individual vials with 1 mL of receptor media and refrigerated overnight. The samples were then warmed to room temperature and sonicated for 15 min, and the supernatant was filtered using 0.22 μm syringe filters into HPLC vials for the assay. The amount of drug (µg) retained in the skin was normalized to 1 g of skin.

### 2.8. High-Performance Liquid Chromatography (HPLC) Analysis

The HPLC system equipped with an Alliance 2695 Separation module and a 2998 PDA-UV detector (Waters Corporation, Milford, MA) was used in this study. The system was interfaced with a workstation loaded with Empower 3 software. The chromatographic separation was performed on a reversed-phase Phenomenex, Luna^®^ C18 Column (5 μm, 250 × 4.6 mm). The mobile phase consisted of 0.3% formic acid in water (solvent A) and acetonitrile (solvent B) in 80:20 ratio. Samples were eluted at an isocratic flow rate of 1 mL/min at ambient temperature. The absorbance wavelength was 254 nm and the run time was 30 min.

### 2.9. Measurement of Cytotoxicity by MTT Assay

For cytotoxicity assessment, B16BL6 cells were cultured in flat-bottom 96-well plates for 24 h. The cell density in the wells was 5000 cells/well. The cells received treatments for 24 h. After 24 h, 10 μL of 3-[4,5-dimethylthiazol-2-yl]-2,5-diphenyl tetrazolium bromide (MTT) was added to each well and the cells were incubated at 37 °C for an additional 2 h. Finally, the medium was aspirated and 100 μL dimethylsulfoxide (DMSO) was added to each well to solubilize the dye remaining in the plates. The absorbance was measured using a microplate reader (Spectramax M5, molecular devices, Sunnyvale, CA, USA) at 544 nm. All experiments were run in triplicate and expressed as mean ± standard error of mean (SEM).

### 2.10. Live/Dead Cell Staining

Fluorescein diacetate (FDA) and propidium iodide (PI) dyes were used to assess cell cytotoxicity after DAB incubation at a concentration of 50 µg/mL. Cells were incubated with 5 mg/mL FDA and 2 mg/mL propidium iodide (PI) for 4 to 5 min at 37 °C to stain live and dead cells, respectively. FDA. is a cell-permeable compound and emits a green fluorescence when it is cleaved by esterases. FDA is used to measure both enzymatic activity and cell membrane integrity [24]. PI is a nuclear staining dye which is impermeable through a viable cell membrane. However, in the compromised cell membrane of dead cells, PI enters the cells and binds to DNA in the nucleus and emits a red fluorescence [25].

### 2.11. Statistical Analysis

All results are presented as mean ± standard error of the mean (SEM). Statistical analysis was performed using Graph Pad Prism (version 9, Graphpad Software, San Diego, CA, USA). All data were subjected to one-way analysis of variance (ANOVA) followed by Turkey–Kramer multiple comparisons test to determine the statistical levels of significance. Mean differences with *p* value < 0.05 were considered statistically significant.

## 3. Results

### 3.1. Preparation and Characterization of Microemulsions

Table 1 details the compositions and characteristics of the investigated microemulsion formulations. The composition of the microemulsions was optimized from different formulation compositions from the ternary phase diagrams with physically stable microemulsion regions. Figure 1a–d shows the pseudo-ternary phase diagrams of the microemulsion system. The microemulsion systems containing MCT oil as an oil phase and Solutol^®^ HS 15 and ethanol as a surfactant mixture demonstrated a small and narrow microemulsion region in the phase diagram (Figure 1a). Similarly, the microemulsion systems containing MCT as an oil phase and Solutol^®^ HS 15 and propylene glycol (in 1:1 and 1:2 ratios) as surfactant mixture also demonstrated a small and narrow microemulsion region in the phase diagram (Figure 1b,c). The composition of microemulsion system containing MCT as an oil phase, lecithin and Solutol^®^ HS 15 as a surfactant mixture (S), and propylene glycol as co-surfactant (co-S) demonstrated a large microemulsion region of the phase diagram. Hence, we have selected this composition with the additional surfactant component lecithin in the formulation. The formulation compositions for F1 and F2 were based on the stable ME region in terms of the physical stability of the emulsions; upon absorption of water, the formulations do not convert to regular emulsions. Hence, F1 and F2 were designed based on the location of the stable microemulsion region of the phase diagram. The physical characteristics of the microemulsions are shown in Table 1. The particle size of the formulations was around 50 nm with a narrow size distribution range as indicated by the smaller PDI values below 0.2 (Appendix A, Appendix A). As evidenced by the data, there was no change in the droplet size or transparency when DAB was loaded to the microemulsion formulation. The pH of the formulations was between 5 and 7, and the viscosity of the formulations was between 500 and 600 mPa-s, which are within the acceptable range for dermatological use.

### 3.2. In Vitro Release and Skin Permeation Studies

Figure 2 shows the DAB release from microemulsions. The formulations F1 and F2 had a similar drug release pattern. A steady increase in DAB as a function of time was observed. The in vitro release profiles of the formulations follow Higuchi release kinetics based on the best fit and higher r^2^ values from the kinetic data analysis (Appendix A).

The skin permeation studies on the microemulsion formulations across intact skin showed no detectable levels of DAB in the receptor phase. However, the drug was significantly retained in the skin after a 24-h equilibration of the formulations with the skin in the Franz diffusion cells.

Figure 3 shows the permeation data of the formulations across tape-stripped skin. As expected, DAB permeated across the skin after tape-stripping. The maximum cumulative amount permeated in 24 h was seen with microemulsion formulations F1 (437 ± 22 µg/cm^2^) and F2 (351 ± 24 µg/cm^2^), whereas the permeation for the control was 43.6 ± 28.8 µg/cm^2^.

The steady-state flux of F1 (16.81 ± 2.75 μg/cm^2^/h) and F2 (12.71 ± 2.75 μg/cm^2^/h) was 8.6- and 6.5-fold higher compared to the control (MCT oil, 1.94 ± 0.73 μg/cm^2^/h). In the intact skin, F2 formulation (26.62 ± 2.05 μg/gm) showed significantly higher DAB levels in the skin compared to MCT oil (9.68 ± 0.67 μg/gm) or F1 (6.59 ± 1.13 μg/gm). However, in the tape-stripped skin, the skin content of DAB was highest for F1 (628 ± 43 μg/gm) followed by F2 (546 ± 57 μg/gm) when compared with control (150 ± 62 μg/gm) (Figure 4).

### 3.3. Cell Cytotoxicity Assays

The percentage of cytotoxicity in B16BL6 cells after 24 h treatment is shown in Figure 5 and Figure 6 (DMEM was used as untreated control). The blank microemulsions (negative controls) did not inhibit cell proliferation. As shown by the data in Figure 5, both microemulsions (F1 and F2) loaded with DAB significantly inhibited cell proliferation compared to the treated control (DAB dissolved in MCT oil) (Figure 5). The F1 and F2 formulations showed significantly higher cytotoxicity toward B16BL6 cell lines compared to the treated control (*p* < 0.001). The half-maximal inhibitory concentrations (IC_50_) of F1, F2, and MCT formulations against B16BL6 cells were calculated to be 1 µg/mL, 10 µg/mL, and 50 µg/mL, respectively.

The qualitative comparison on live/dead cell staining demonstrated that the microemulsion treated cells showed lower live signal, while the control (no DAB) showed more live and less dead signal (Figure 7). In addition, the morphology of cells with DAB-MEs was more spherical, which inhibited growth and probably damaged cells. Microemulsion showed more cytotoxicity and live/dead signal compared to the treated control.

## 4. Discussion

Diacetyl Boldine is an antioxidant that is commonly used in antiaging products and cosmetics for reducing pigmentation and melasma. Boldine, the parent alkaloid, has shown cytotoxicity in many cancers [7,8,9,10], but the anti-melanoma effect is not reported. We designed microemulsion formulations for DAB based on the optimal solubility of DAB in the formulation components for enhanced skin delivery and cytotoxicity in melanoma cells. The formulation composition was optimized from the data obtained from pseudo-ternary phase diagrams of the microemulsion system. The MCT oil with Solutol^®^ HS 15 and PG or ethanol formulations without lecithin yielded a narrow region for the existence of microemulsion (Figure 1a–c). When lecithin was included, the microemulsion area was improved. The formulations F1 and F2 were designed based on the location of the stable microemulsion region of the phase diagram (Figure 1d). The particle size was around 50 nm with a narrow PDI suggesting a stable microemulsion for topical delivery (Appendix A). These formulations were found to be miscible with water in all proportions without yielding any cloudiness confirming the high physical stability of the microemulsion system. The optimal dermatological pH and the lower viscosity of the formulations (Table 1) enable easy spreadability of the microemulsion formulations on the skin.

The in vitro release study was conducted to show that the drug is readily available for skin permeation. Both formulations showed a similar drug release pattern. The in vitro release profiles of the formulations suggest matrix diffusion kinetics (Q versus square root of time) according to the Higuchi’s model (Figure 2). The release data show a higher r^2^ value for Higuchi’s equation compared to zero-order, first-order, Hixson–Crowell, and Korsmeyer–Peppas models. (Appendix A). The drug release from microemulsions followed Higuchi kinetics for curcumin [29] and fluticasone or levocetrizine [30]. This Higuchi model is based on Fick’s law of diffusion, which provides a linear relationship between the amount of drug released versus square root of time. In this model, the formulation drug release is regulated by a diffusion-controlled mechanism where the drug diffusion across a defined area is proportional to the concentration gradient among the diffusion layers [31]. Many factors affect the diffusion such as the drug’s concentration gradient between the donor and receptor cells, aqueous solubility, and the partition coefficient between the release membrane and the receiver medium. The diffusion gradient across the cellulose membrane can be dependent on the drug concentration in the internal phase of the microemulsion. Due to a higher drug concentration in the oil phase, misalignment of the surfactant film occurs, leading to drug diffusion and transfer to the external phase of the microemulsion [30]. However, zero-order kinetics were obtained for tretinoin and metronidazole microemulsion formulations [32,33]. These studies did not provide information on the type of dialysis or release membrane in the experiments. The membrane might offer resistance to diffusion leading to non-Higuchi kinetics.

The DAB levels below detectable levels of permeation across intact skin suggest that the stratum corneum is the major barrier for DAB permeation. However, DAB levels were detected in the skin for all the formulations. It is has been well established that the tape-stripped skin shows dramatic increases in the skin permeation of moderately water-soluble compounds such as caffeine [34], crisaborole [35] and vancomycin [36]. Likewise, diacetyl boldine permeation and skin retention across tape-stripped skin (with no stratum corneum) was very high compared to intact skin (Figure 3 and Figure 4). The F2 formulation showed significantly higher DAB levels in the intact skin versus F1 suggests increased oil and reduced water content facilitates the skin retention of DAB. The higher DAB skin retention by F2 versus MCT-DAB (DAB dissolved in MCT) suggests that microemulsion greatly facilitates the accumulation of the drug in the skin. Microemulsions have been used to improve skin permeation for many drugs. Paolino, et al. [37] reported that ketoprofen microemulsion showed higher permeation than conventional formulations (emulsion and gel). Furthermore, a microemulsion improved the diclofenac skin permeation compared to a liposome formulation [38]. The transdermal flux for azelaic acid microemulsion was significantly higher than the drug solution [39].

To understand the mechanism of skin permeation enhancement by microemulsion formulations, we conducted permeation studies across the tape-stripped skin (Figure 4). As expected, the permeation across the skin after the tape-stripping was high. The tape-stripping technique removes the stratum corneum, enabling the drug to cross the skin. This study further proves that the stratum corneum is a significant barrier to skin permeation of DAB. The flux of DAB by both microemulsions (F1 and F2) was substantially increased as compared to DAB-MCT (oil solution). Accordingly, the DAB skin content was high for both formulations versus control, proving that microemulsion is a better vehicle for the delivery of DAB (Figure 4).

Stratum corneum is the rate-limiting step for transdermal delivery for many drugs. By removing the stratum corneum, DAB permeation and skin retention was enhanced for microemulsions as well as DAB-MCT (solution). Similar results were obtained for different drugs. Arima et al. have reported that the flux of 4-biphenylyl acetate, which is a prodrug of the non-steroidal anti-inflammatory drug, across tape-stripped skin was greater than that through full-thickness skin [40]. Furthermore, the tetramethylpyrazine microemulsion showed a two-fold increase in drug permeation across skin treated with microneedles compared with formulation alone [41]. A combination of microneedle with microemulsion to deliver celecoxib increased skin permeation more than microemulsion alone [42]. All these studies suggest that drug permeation is much higher, especially for hydrophilic drugs, for barrier-compromised skin [43].

The effect of microemulsions in the absence and presence of DAB on the cytotoxicity was determined in comparison to DAB dissolved in MCT oil (treated control). The blank microemulsions (ME1 and ME2) at the highest concentration did not show any cytotoxicity (Figure 6). Likewise, the MCT oil vehicle did not show any cytotoxicity. MCT oil consists of caprylic and capric triglycerides which are obtained from natural and synthetic origin [44]. It is widely used in several pharmaceutical products. MCT oil (Miglyol 810N), super refined propylene glycol, and phosphatidyl choline are listed as “Generally Recognized as Safe” ingredients according to the US FDA. They have been widely used as solvents, suspending agents, emulsifying agents, and therapeutic agents in various topical creams, microemulsions, and ointments. Fiume et al. conducted extensive safety assessment on phospholipids such as lecithin and showed that even higher concentrations of lecithin are nontoxic and did not lead to skin irritation [45]. Similarly, previous studies have shown that inclusion of a high concentration of propylene glycol did not induce any visible irritation or toxicity after the application of microemulsion for 7 days on the skin of rabbits [46]. Similarly, Solutol HS 15 has emerged as a great alternative to Cremophor EL and is regarded as a nontoxic and nonirritant excipient. Even higher concentrations up to 50%*v*/*v* were reported to be used in the preparations of diclofenac and vitamin K1 formulations [47]. Therefore, in the current study, excipients which are nontoxic and are not expected to show low human skin irritancy were selected for the preparation of microemulsions.

The IC_50_ of F1, and F2 were 1 and 10 µg/mL, respectively, whereas the IC_50_ of DAB-MCT oil solution was 50 µg/mL, suggesting the microemulsion formulation was a more effective vehicle for DAB. The amount of DAB accumulated in the intact skin for formulations F1 and F2 under infinite dose conditions was high enough for inhibiting the cell proliferation of B16BL6 cells compared to that of control. Additional in vivo studies are warranted to determine the efficacy of the formulations. Boldine, due to its free radical scavenging properties, influences cell proliferation, survival, differentiation and metabolism. In cancer cells, it can scavenge high amounts of H_2_O_2_ that stimulates proliferation. Studies have shown that, at 10–50 µM concentrations, boldine exhibits intense free radical scavenging properties [48,49]. The live/dead cell staining results indicate that the microemulsions were more potent in inducing cell death (Figure 7). This was due to high homogeneity and nanoscale, which led to endocytosis and increased drug uptake. The cell membrane is impermeable to large particles; particles 10–30 nm can easily cross cell membrane [50]. The amphiphilic properties of surfactant and the small droplet size of the microemulsion have facilitated drug diffusion into the cells and cell endocytosis for several drugs such as etoposide and coix seed oil [51], caffeic acid phenethyl ester [52], and tamoxifen [53]. Because the droplet size of the microemulsion was ~50 nm, similar mechanisms have likely contributed to enhanced cellular uptake of DAB and enhanced drug anti-proliferation activities. Many studies have shown improved anti-proliferative activity of drugs after nanoencapsulation. For example, myricetin microemulsion exhibited a significant increase of anti-proliferative activity against liver cancer cells compared to free myricetin [54]. A significantly better anti-proliferation effect was found on colon cancer cells for tangeretin prepared in emulsion with lecithin and medium chain triacylglycerol, compared to tangeretin solution [55]. Curcumin loaded in nanostructured lipid carriers inhibit the growth of ovarian cancer more than curcumin solution [56].

## 5. Conclusions

The data from pseudo-ternary phase diagrams suggest that the lecithin-based formulations can provide physically stable microemulsions as known from the broader ME region. The formulations showed a very small mean globule size of around 50 nm with a narrow distribution, as known from the polydispersity index of less than 0.2. All formulations followed the Higuchi model based on the R^2^ value of release kinetic analysis. The skin permeation data showed that the F1 and F2 microemulsion formulations provided significantly higher skin retention levels as compared with the DAB solution in MCT oil. Besides, the formulations showed significantly higher cytotoxicity toward B16BL6 cell lines compared to the control formulation (*p* < 0.001). The cytotoxicity studies show that F1 and F2 formulations show a significantly higher cytotoxicity for the DAB microemulsions compared to the control formulation, DAB in MCT oil (*p* < 0.001). The amount of DAB accumulated in the intact skin for formulations F1 and F2 under infinite dose conditions was high enough to inhibit the cell proliferation of B16BL6 cells. Additional in vivo studies are warranted to know the efficacy of the formulations. Overall, the results of the present study suggest that microemulsion could be a promising vehicle for the topical administration of DAB for its chemo protection properties against melanoma.

## Figures and Tables

**Figure 1 pharmaceutics-15-00901-f001:**
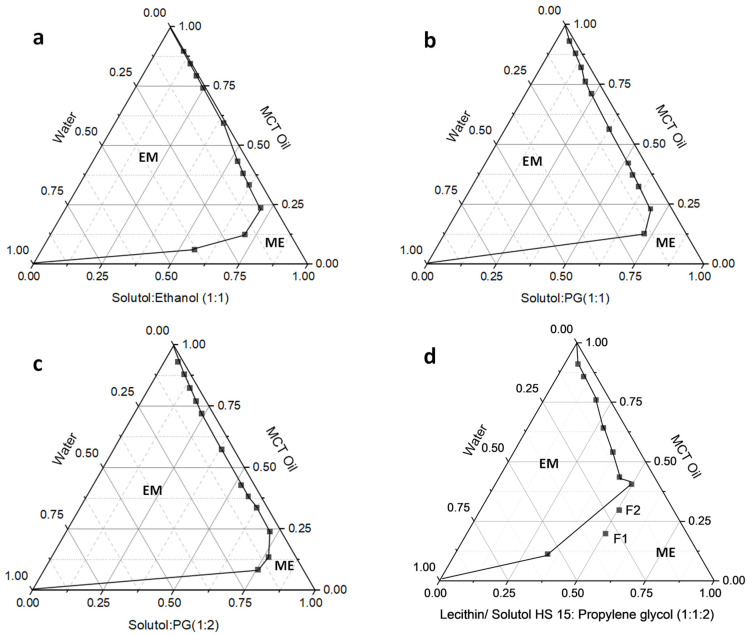
Pseudo-ternary phase diagrams of MCT Oil, Lecithin, Solutol^®^ HS 15, Propylene glycol, Water system; EM—emulsion, ME—microemulsion.

**Figure 2 pharmaceutics-15-00901-f002:**
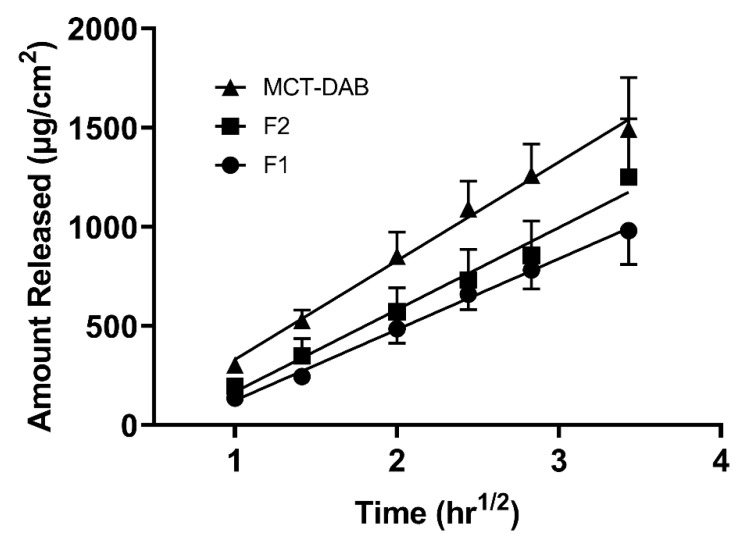
In vitro release of DAB microemulsion formulations across a dialysis membrane. All data are expressed as mean ± standard error, n = 3.

**Figure 3 pharmaceutics-15-00901-f003:**
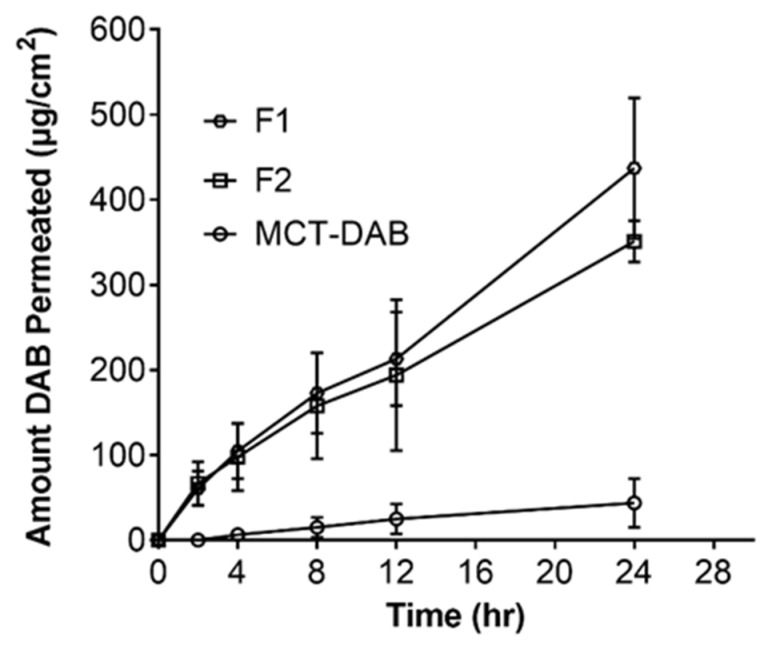
Skin permeation and flux of DAB from microemulsions after tape-stripping of human cadaver skin. All data are expressed as mean ± standard error, n = 3.

**Figure 4 pharmaceutics-15-00901-f004:**
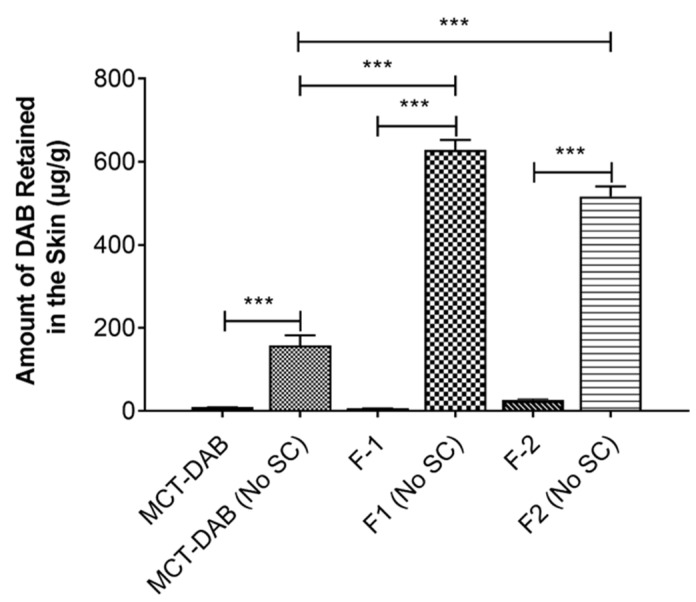
Effect of stratum corneum tape-stripping on the skin retention of DAB in human cadaver skin. All data are expressed as mean ± standard error, n = 3. *** *p* < 0.001 as compared with MCT-DAB.

**Figure 5 pharmaceutics-15-00901-f005:**
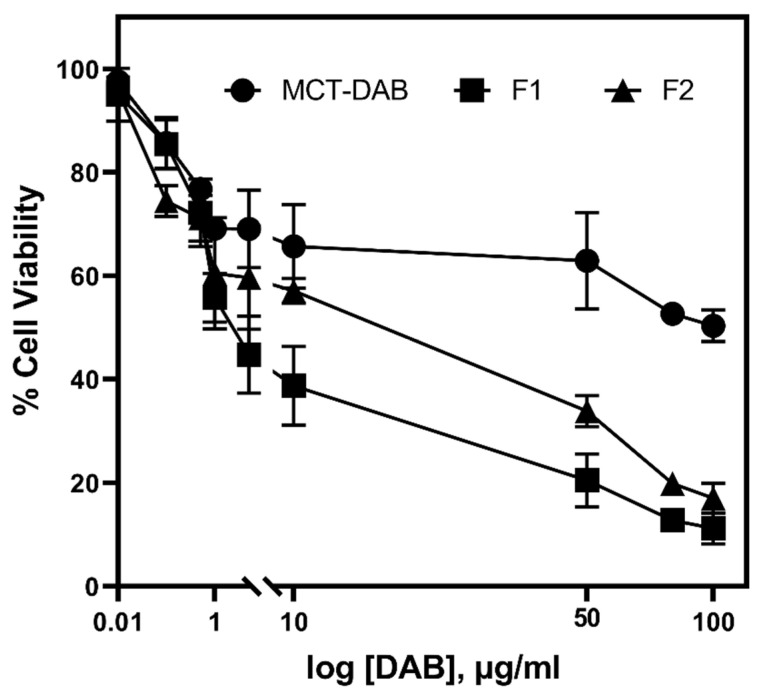
Cytotoxic effect of DAB on the B16BL6 cell line. Cells were incubated with various concentrations for 24 h. All data are expressed as mean percentages to untreated control cells (n = 3).

**Figure 6 pharmaceutics-15-00901-f006:**
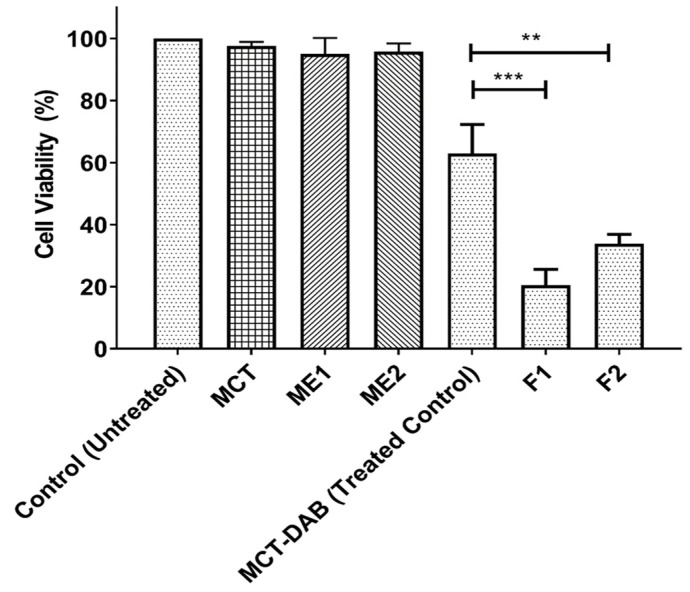
In vitro cytotoxicity of formulations with or without DAB in B16BL6 melanoma cell line at 50 µg/mL DAB concentration. All data are expressed as mean ± standard error, n = 3. *** *p* < 0.001 when F1 compared MCT-DAB. ** *p* < 0.01 when F2 compared to MCT-DAB.

**Figure 7 pharmaceutics-15-00901-f007:**
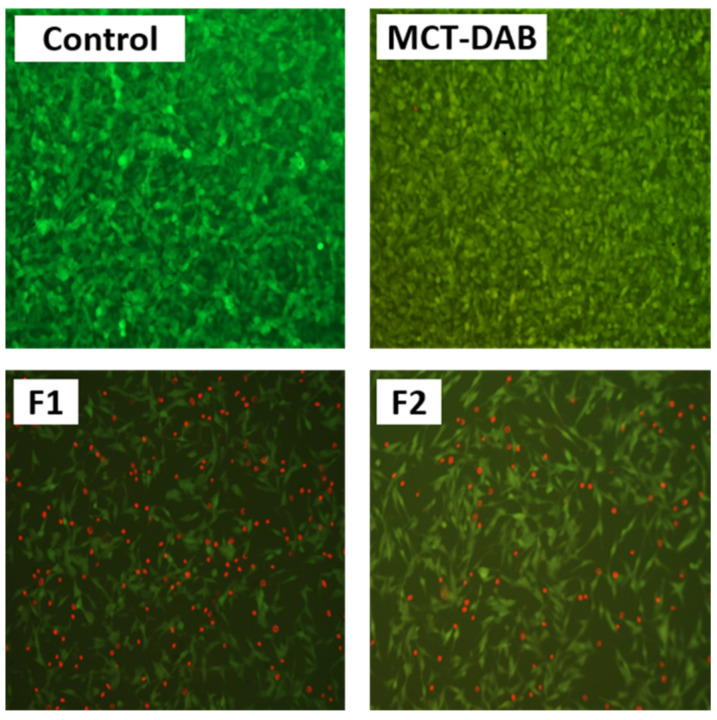
Live/dead cell staining of melanoma cell line. Cells were treated with 50 µg/mL for 24 h. Live cells appear green, while dead cells appear red.

**Table 1 pharmaceutics-15-00901-t001:** Composition and characteristics of microemulsion formulations.

Formulation Code	ME1	ME2	F1	F2
**Composition (%*w/w*)**				
Diacetyl Boldine	0%	0%	1%	1%
MCT Oil	20%	30%	20%	30%
Lecithin	13%	13%	13%	13%
Solutol^®^ HS 15	13%	13%	13%	13%
Propylene Glycol	25%	25%	25%	25%
Water	29%	19%	28%	18%
**Characteristics**				
pH	5.47 ± 0.20	5.40 ± 0.16	6.23 ± 0.03	6.20 ± 0.04
Particle size (nm)	49.77 ± 2.79	35.19 ± 0.24	54.96 ± 0.47	33.48 ± 0.28
Polydispersity Index (P.D.I.)	0.151 ± 0.03	0.085 ± 0.01	0.146 ± 0.003	0.06 ± 0.002
Viscosity (at 1 rpm, cP)	510.13 ± 3.25	536.26 ± 3.25	506 ± 3.30	551.56 ± 6.81

## Data Availability

Not applicable.

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
