# Peer review of "Topical Delivery of Diacetyl Boldine in a Microemulsion Formulation for Chemoprotection against Melanoma"

_pharmaceutics, 2023, doi:10.3390/pharmaceutics15030901_

Round 1

Reviewer 1 Report

The present research manuscript titled “Topical Delivery of Diacetyl Boldine Microemulsion as a Chemoprotective Agent against Melanoma” is very concise and well-written. The research design is good and the obtained results are interesting. However, many flaws are present manuscript. This manuscript needs critical revision. My comments are as follows.

Comment 1. The title of the manuscript is illogical. The drug/phytochemical can be a chemoprotective agent but microemulsion can’t. Therefore, the authors should revise the title.

Comment 2. The introduction section is poor. The authors should discuss melanoma, challenges in its treatment, statistics, etc. in the introduction section.

Comment 3. The authors should incorporate the images of particle size, and zeta potential in the manuscript.

Comment 4. What about the morphology of the particles? The authors should examine the morphology of the particles by TEM and report in the manuscript.

Comment 5. Section 3.1: The results should be mentioned with their standard deviation. Kindly revise this section.

Comment 6. The in vitro release profiles follow zero-order release kinetics based on the R2 value from the data analysis. This statement is completely wrong. The authors should fit the release data in different mathematical models to choose the best-fitted model to explain the mechanism of drug release from microemulsion and report in the manuscript.

Comment 7. The researcher should perform the cellular uptake study to verify the cytotoxicity results.

Comment 8. The conclusion section is very poor. The authors should revise this section and report major findings and make a conclusion on the basis of obtained results in at least 100-150 words.

Reviewer 2 Report

The authors describe the preparation and characterization of a microemulsion containing diacetyl boldine. Based on some preliminary results, the authors conclude that the prepared microemulsions promote the uptake of diacetyl boldine (DAB) into cells, increasing its cytotoxic effect. The topic of the manuscript is interesting, but the manuscript has serious shortcomings. More measurement results and a much deeper evaluation of the results are needed for the manuscript to be accepted as a scientific publication.

–  I think the number of measured data is insufficient to make phase diagrams, especially for Figure 1c. What types of (micro)emulsions exist in the marked regions?
– What is the role of Figures 1 a,b, and c? These results are not included in the discussion, not even among the results. There is also a lack of scientific justification for the selection of the tested samples (F1, F2).
– Figure 2: Based on the data points, I hardly believe in zero-order kinetics. The curves appear to be exponential. The points of the diagrams should not be connected by lines from point to point. Calculate the best-fitting curve!
– Figure 4b is not discussed.
– Figure 6 is not discussed. The x-axis should be converted to logarithmic form.
– Did you examine a non-cancerous cell line? What is the toxicity of the formulation on healthy cells?

Minor notes:
– Explain abbreviations the first time they are mentioned, e.g., F1, F2, MCT.
– “Several proposed mechanisms explain the penetration effect of microemulsions such as, increase in skin hydration, nano-droplet size and enhanced surface area.” These are not mechanisms.
– Please provide more details about medium-chain triglyceride (MCT) oil.
– The name of the applied post hos test is “Tukey-Kramer post hoc test”.
– In the case of microemulsions, the term “droplet size” is recommended instead of “particle size”.
– Instead of cP use mPas.
– One of the axis labels in Figure 1a is missing.
– Be more careful with similar claims: “The F2 formulation penetrated and retained in the skin 13-fold higher than the control (DAB-MCT oil)”. You measured only the concentration of DAB.  More precise wording is needed. A 13-fold higher concentration (based on Figure 3.)?

Author Response

The authors' responses to the comments attached

Reviewer 3 Report

Some minor remarks:

Line 50: “Approximately 60% of 50 commercially available anticancer drugs are plant-derived and from natural resources [2].” Please find more recent publications.

Line 89: “Besides, there are no published studies on the topical delivery of DAB across human or animal skin” See, for example: J. Cosmet. Dermatol. 2016, 15(2):131-44. doi: 10.1111/jocd.12201.

Line 145: “It was collected from a single donor within 8 h of death and frozen in 50% glycerol in normal saline at 80° C until use.” May be authors mean 80° K or -80° C?

Discussion: please discuss the molecular mechanism of DAB cytotoxicity if it is known. The steady state flux and skin content of DAB was very similar for the microemulsions F1 and F2, but IC50 values differ 10 times, why?

Author Response

(The authors gave the same response as above.)

Round 2

Reviewer 1 Report

I appreciate the efforts made by the researcher. The quality of the manuscript improved significantly after the revision. However, I have a few more queries. My comments are as follows. 

Comment 1. The size of the developed formulation is around 50 nm. Then, why the researchers claimed that this is a microemulsion? Kindly explain.

Comment 2. Page 9, lines 284 & 285: The results look like a fabricated one. Kindly verify and then mention it in the manuscript. 

Author Response

Please see attached response.
